

# Different ecological demands shape differences in population structure and behaviour among the two generations of the small pearl-bordered fritillary

Ann-Kathrin Sing[1,2], Laura Guderjan[1,3], Klara Lemke[1,4], Martin Wiemers[1], Thomas Schmitt[1,5] and Martin Wendt[1,6]

[1] Senckenberg German Entomological Institute, Müncheberg, Germany
[2] Institute of Earth and Environmental Sciences, Albert-Ludwigs-University Freiburg, Freiburg, Germany
[3] Institute of Geoecology, Technische Universität Braunschweig, Braunschweig, Germany
[4] Faculty of Natural and Environmental Sciences, Zittau/Görlitz University of Applied Sciences, Zittau/Görlitz, Germany
[5] Institute of Biochemistry and Biology, University of Potsdam, Potsdam, Germany
[6] Leibniz Centre for Agricultural Landscape Research, Müncheberg, Germany

Corresponding author
Thomas Schmitt,
Thomas.Schmitt@senckenberg.de

## ABSTRACT

The population structure and behaviour of univoltine butterfly species have been studied intensively. However, much less is known about bivoltine species. In particular, in-depth studies of the differences in population structure, behaviour, and ecology between these two generations are largely lacking. Therefore, we here present a mark-release-recapture study of two successive generations of the fritillary butterfly *Boloria selene* performed in eastern Brandenburg (Germany). We revealed intersexual and intergenerational differences regarding behaviour, dispersal, population characteristics, and protandry. The observed population densities were higher in the second generation. The flight activity of females decreased in the second generation, but remained unchanged in males. This was further supported by the rate of wing decay. The first generation displayed a linear correlation between wing decay and passed time in both sexes, whereas the linear correlation was lost in second-generation females. The proportion of resting individuals in both sexes increased in the second generation, as well as the number of nectaring females. The choice of plant genera used for nectaring seems to be more specialised in the first and more opportunistic in the second generation. The average flight distances were generally higher for females than for males and overall higher in the first generation. Predictions of long-distance movements based on the inverse power function were also generally higher in females than in males but lower in the first generation. Additionally, we found protandry only in the first but not in the second generation, which might correlate with the different developmental pathways of the two generations. These remarkable differences between both generations might reflect an adaptation to the different ecological demands during the flight season and the different tasks they have, *i.e.*, growth in the spring season; dispersal and colonisation of new habitats during the summer season.

## INTRODUCTION

A unique combination of attributes makes butterflies promising biodiversity indicators. Butterflies have short life cycles, they react sensitively to changes in their habitat, they can breed in small habitat patches, and they represent a wide range of different terrestrial ecosystems (*Thomas et al., 2004*; *Thomas, 2005*; *van Swaay, Warren & Loïs, 2006*; *Settele et al., 2008*). Most butterfly species have special needs regarding habitat quality, nectar resources, microclimate, and the availability of larval host plants (*Fartmann & Hermann, 2006*; *Münsch, Helbing & Fartmann, 2019*). They therefore can be seen on a generalist-specialist continuum depending on their degree of specialisation and the association with different life history traits (*Dapporto & Dennis, 2013*). Specialists are characterised by a narrow feeding niche, small body size, high population density, low dispersal propensity, few generations per year and long developmental time (*Koh, Sodhi & Brook, 2004*; *Komonen et al., 2004*; *Tscharntke et al., 2005*; *Bartonova, Benes & Konvicka, 2014*). Specialised species might suffer the most from habitat degradation due to their use of scarce habitats and a poorer capacity to colonise new patches (*Koh, Sodhi & Brook, 2004*; *Cardoso et al., 2020*; *Modin & Öckinger, 2020*).

As ectothermic individuals, butterflies depend on environmental cues that control and influence important life-history traits. This is particularly important in temperate zones with their seasonality, forcing animals to evolve different adaptations to deal with these varying conditions. Thus, climatic variability can lead to changes in the length of the activity period, shifts of the activity period to earlier times of the year and changes in generation numbers (*Yamamura & Kiritani, 1998*; *Høye et al., 2014*; *Zografou et al., 2021*). As a consequence, the number of generations varies with temperature and photoperiod and therefore with latitude and altitude; populations of the same species thus can have two or more generations at lower altitudes or latitudes but the second can be partial or there can be just one at higher elevations or latitudes (*Schweizerischer Bund für Naturschutz, 1987*; *Altermatt, 2010*). The generations in plurivoltine species, meaning species with more than one generation, are characterised by different developmental pathways: either direct development within the same season or interrupted by diapause during the cold season (*Wiklund & Solbreck, 1982*). Consequently, the subsequent generations of plurivoltine butterflies are subject to different seasonal selection regimes what can lead to seasonal polyphenism (*Komata & Sota, 2017*; *Esperk & Tammaru, 2021*).

Further, butterfly species with separate generations frequently show protandrous emergence (*Nève & Singer, 2008*). This strategy enhances reproductive success for both males and females because males need some time to enter the reproductive phase so that protandry ensures the fertilisation of females (*Fagerström & Wiklund, 1982*; *Zonneveld, 1992*). Experimental studies revealed that protandry can persist under different temperatures during larval development (*Fischer & Fiedler, 2000*) and under different

developmental pathways (*Fischer & Fiedler, 2001a*; *Karl & Fischer, 2008*). However, protandry is often reduced or absent in very harsh environments like high mountain habitats (*Junker et al., 2010*) where in case of protandry weather turbulences might kill males prior to the emergence of females hereby potentially eradicating entire populations. However, although mostly being a beneficial trait, maintaining protandry over subsequent generations might not be feasible. Thus, reduction or complete absence of protandry in the second generation already has been demonstrated (*Wiklund, Nylin & Forsberg, 1991*; *Komata & Sota, 2017*).

Among life history traits, dispersal also has far reaching fitness consequences, as it actively allows the evasion of stressors and the establishment of new (meta)populations. Dispersal is distinguished from everyday movement by a large net-displacement through the matrix into a new habitat (*Schtickzelle et al., 2007*; *Korösi et al., 2008*) and the possibility of individuals to increase their inclusive fitness by displacing (*Bowler & Benton, 2005*; *Schtickzelle et al., 2007*; *Hovestadt & Nieminen, 2009*). This behaviour is promoted by high population density, bad habitat quality, small patch size and predation and can reduce kin competition and inbreeding (*Bowler & Benton, 2005*; *Hovestadt & Nieminen, 2009*). The costs of dispersal are delay in reproduction, reduced longevity, and the risk to die or failing to reproduce (*Hovestadt & Nieminen, 2009*).

The aim of our study is to examine how important traits like protandry, behaviour, and dispersal vary between subsequent generations of butterflies. Since variability in mobility is already reported for other species (*Plazio & Nowicki, 2021*), a seasonal polyphenism in dispersal and flight behaviour might exist as a consequence. While for protandry, laboratory experiments exist for comparisons of the different developmental pathways, it is important to analyse whether these changes also occur under natural conditions in the field. Studies on plurivoltine species and comparisons among different phenotypes of the same species, *i.e.,* the successive generations, have the advantage that the groups can be considered to have the same physiological constraints (*Wiklund & Solbreck, 1982*) and are genetically identical so that differences necessarily are caused by phenotypic plasticity (*Esperk & Tammaru, 2021*).

While numerous specialised univoltine butterfly species are already well-studied, there is a research-gap regarding field studies on bivoltine species and comparisons between their different generations (*Fric & Konvička, 2000*; *Plazio & Nowicki, 2021*; *Hajkova et al., 2023*). In our study, we therefore selected the small pearl-bordered fritillary *Boloria selene* (Nymphalidae) as a model species. This species inhabits open grassland habitats, mostly damp to wet meadows and fens (*Ebert & Rennwald, 1991*). We performed a mark-release-recapture (MRR) study embracing two subsequent generations in wet meadows in eastern Brandenburg (Germany) to investigate the following three research questions:

(1). Does protandry occur in both generations and is there a difference in the length of males' earlier emergence?

(2). Are there differences between the two generations regarding dispersal and general behaviour of the individuals?

(3). If existing, what are the causes of the differences between the two generations?

By answering these questions, we intend to better understand the task of each generation along the year and how they are adapted to cope with the different environmental conditions prevailing during the respective time windows. Hereby, we hope to disentangle the evolutionary pressures acting on such bivoltine species and their adaptation strategies.

## MATERIALS & METHODS

### Study species *Boloria selene*

The small pearl-bordered fritillary *Boloria selene* (Denis & Schiffermüller, 1775) belongs to the family Nymphalidae, subfamily Heliconiinae. *Boloria selene* is widely distributed in Eurasia at elevations from 0 to 2,200 m asl (*Tuzov & Bozano, 2006*; *Kudrna et al., 2011*). The species inhabits nutrient-poor open landscapes, damp to wet meadows, fens and fen edges with abundance of flowering plants but also forest paths and clearings (*Settele et al., 2008*; *Settele et al., 2015*). The host plants of the caterpillars are different violet species (*Viola* spec.), in Germany mostly the marsh violet (*Viola palustris*) (*Ebert & Rennwald, 1991*; *Gelbrecht et al., 2016*) but also *V. canina*, *V. riviniana* and *V. hirta* (*Tolman & Lewington, 1998*). In the temperate zone, the species is mesophilic (*Gelbrecht et al., 2016*) and bivoltine with the second generation sometimes just being a partial one (*Settele et al., 2008*); however, it can be univoltine at higher altitudes, *e.g.*, above 1,000 m asl. in the Alps, and in northern Europe (*Schweizerischer Bund für Naturschutz, 1987*). The two subsequent generations are temporally separated; the flight period of the first generation is mid-May to mid-June; the second generation is on the wing from late July to late August (*Ebert & Rennwald, 1991*; *Gelbrecht et al., 2016*). The adults feed on a wide spectrum of flowering plants (*Schweizerischer Bund für Naturschutz, 1987*; *Ebert & Rennwald, 1991*). In a comparison with other *Boloria* species, *Boloria selene* is graded as a generalist species regarding the use of multiple nectar resources as an adult and no known specialisation in other resources and microhabitat structures (*Turlure et al., 2010*; *Turlure et al., 2019*). On the German Red List, *Boloria selene* is listed on the pre-warning list (*Reinhardt & Bolz, 2011*), while it is categorised as endangered on the Red List of Brandenburg (*Gelbrecht et al., 2001*; *Settele et al., 2015*).

### Study site

The study was carried out in eastern Brandenburg (North-East Germany) in the administrative district of Märkisch-Oderland, near the city of Müncheberg (Fig. 1). The annual average temperature is 9.0 °C and the annual average rainfall 563 mm (1981–2010; DWD weather station Müncheberg (*Deutscher Wetterdienst (DWD), 2010*). The study site, the so-called "Gumnitzwiesen" (52°30′N, 14°04′E), is part of the nature reserve "Gumnitz und Großer Schlagenthinsee" and the Natura 2000 site of the same name, which are completely embedded in the nature park "Märkische Schweiz". The study area of approximately 10.5 ha consists mainly of nutrient-poor to nutrient-rich wet meadows of the plant community *Molinion caeruleae* and smaller parts of dry, sandy, calcareous grassland (*Landesamt für Umwelt, 2019*). The area is delimited by a large pine forest and a smaller fen woodland area, a lake at the eastern border and a forest path used for hiking alongside the forest edge (*Landesamt für Umwelt, 2019*). The next known habitats of *Boloria*

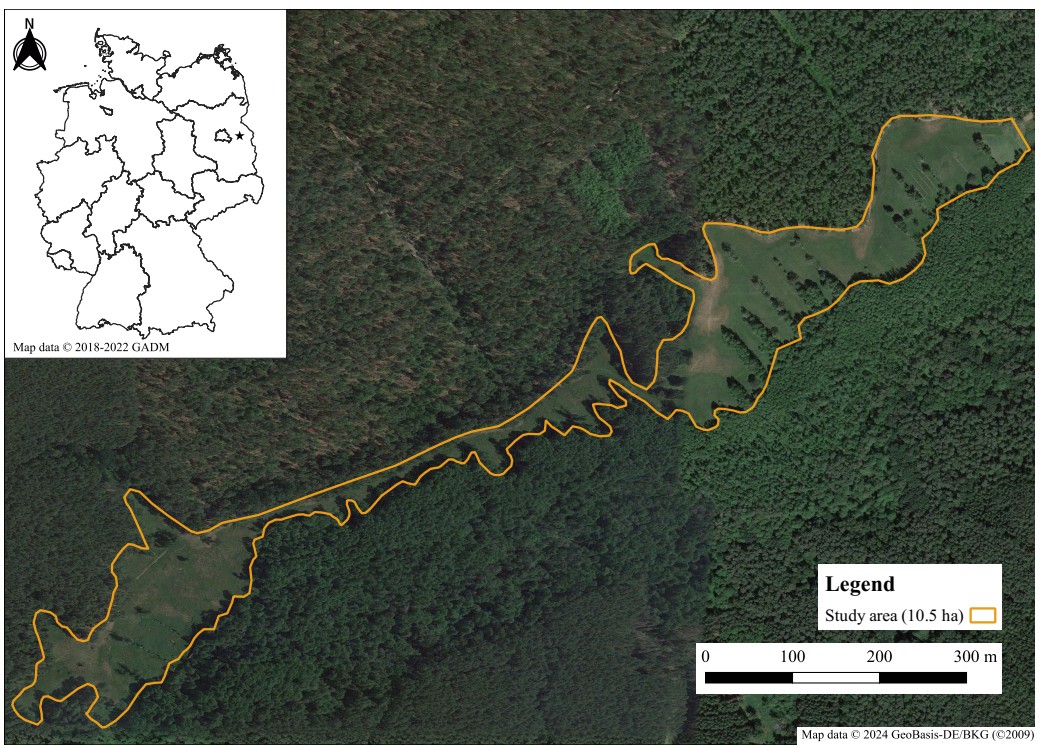

**Figure 1** Map of the study area "Gumnitzwiesen" in Müncheberg, eastern Brandenburg. Map data ©2024 GeoBasis-DE/BKG (©2009); small map: ©2018–2022 GADM.

*selene* are several km away from our study site so that exchange of individuals should be rare or completely missing.

## Data collection

We applied the mark-release-recapture (MRR) method (*Ehrlich & Davidson, 1960*). Butterfly individuals were captured with an insect net, marked with a continuous, unique ID on the undersides of fore- and hindwings with a permanent felt tip pen ("OHpen universal" in black by Stabilo®) and released immediately after capture. The ID consists of a letter that represents the capture day and a running number, starting each day with 1. We took the GPS position of every capture and recapture event using smartphones with the app TourCount© (*Stein, 2019*). For every capture event, we noted the individual's sex, its wing condition and activity prior to capture, the plant species or genus when the individual was feeding, temperature, wind speed (scale: 1–4) and cloudiness (percentage of sky coverage). The wing condition was classified on a scale from 1 to 4 (1: fresh with undamaged wing fringe, 2: missing wing fringe, 3: slightly damaged, 4: heavily damaged) (*Thomas, 1983*; *Zimmermann et al., 2005*).

Data were collected 24 May–20 June 2021 (first generation) and 11 July–13 August 2021 (second generation). Hence, our analyses all rely on data collected during one single year. However, weather conditions were favourable during the entire field work period and the population densities were well manageable so that our data should be well suitable for

representing our study species. Nevertheless, it would be nice to have data from additional years to cross-check our results. The daily number of collectors varied from one to six people, *i.e.,* three people regularly participating, who were sometimes accompanied by up to three students. The whole study area was surveyed using transects. Every cluster of *Viola palustris* was recorded with its GPS position using the app TourCount©.

## Ethics declaration
No butterflies were harmed or killed for this study. Field work was authorised by Landesamt für Umwelt, Gesundheit und Verbraucherschutz, Frankfurt (Oder).

## Data analyses
### Population demography
Daily population sizes were estimated using the POPAN 5.0 module in the program MARK 9.0 (*Cooch & White, 2021*) that is based on the Jolly-Seber model for open populations (*Bellebaum, Köppen & Grajetzky, 2010*). Population sizes were estimated separately for sexes. POPAN estimates the three primary parameters: survival probability (phi), capture probability (p) and the probability of entry into the population (pent). These parameters are tested for different dependencies and may depend on sex (g), on time (t) or be constant. The factor time may either be factorial (t), linear (T) or quadratic ($T^2$), and the relationship with group dependencies might be additive (+) or interactive (x) (*White & Burnham, 1999*; *Junker & Schmitt, 2010*; *Kadlec et al., 2010*). Sampling effort as a covariant of recapture probability was considered with a factor of 1 for each person (unskilled worker 0.5; teams 1) times the hours of sampling. After a Goodness-of-Fit test (option: RELEASE), we calculated the dispersion factor ĉ to adjust the AIC of the models for over- or underdispersion in the data. The best model was chosen using the corrected Akaike Information Criterion (*Sugiura, 1978*; *Hurvich & Tsai, 1989*; *Lebreton et al., 1992*). Modelling was conducted separately for each generation.

### Behaviour, nectar plant preference and wing conditions
Differences between sexes in behaviour and preferences of nectar plant genera were tested with a Chi-squared test of homogeneity; for behaviour, each individual entered only once with its first record to avoid pseudoreplication. The mean wing condition was analysed separately for both sexes to assess the age structure of the population (*Watt et al., 1977*). Days with fewer than five specimens were excluded. The correlation of wing condition with time was tested using Pearson correlation (*Ehl et al., 2017*). The general difference between sexes was tested with Mann–Whitney *U* tests, and the difference between the slopes of the correlations was tested using a multiplicative model with the interaction of time and sex as predictor variables. The within-sex differences between generations were tested in the same way.

### Mobility and movement patterns
Based on GPS data of capture and recapture events, the movement of the individuals was analysed using QGIS 3.18.3 (*QGIS, 2021*). The distances travelled between first and second capture were measured as direct paths and tested for differences between sexes

and between generations. Lifespan movements were measured as the addition of every single direct distance moved, from first to last capture. For comparisons between sexes and generations, we used Mann–Whitney $U$ tests. We calculated the inverse cumulative proportion of individuals that reached certain distance classes with intervals of 20, 30, and 50 m, respectively. For predictions of long-distance movements, we fitted the data against two commonly applied mathematical models, the negative exponential function (NEF) and the inverse power function (IPF), separately for both sexes and generations (*Hill, Thomas & Lewis, 1996*; *Baguette, 2003*; *Zimmermann et al., 2005*). The equation of the NEF is $I_{NEF} = ae^{-kD}$ and respective $\ln I = \ln a\ kD$, for the relative proportion of individuals ($I$) moving to distance $D$. The parameter $k$ of the equation is a dispersal constant, and the parameter $a$ stands for the intercept of the regression. The equation for $I$ within the IPF is $I_{IPF} = cD^{-n}$ and respective $\ln I = \ln c - n\ (\ln D)$, with $c$ as a scaling constant and $n$ as slope, describing the effect of distance on dispersal (*Fric & Konvička, 2007*; *Junker & Schmitt, 2010*; *Junker & Schmitt, 2010*). The best model and interval size were chosen based on the stability index $R^2$ of the fitted models. All statistical tests were calculated in RStudio version 1.4.1103 with the package "stats" (*RStudio Team, 2021*).

## RESULTS

In the first generation (21 capture days; 24 May–20 June 2021), 243 individuals were marked (129 males, 114 females) of which 176 were recaptured at least once (102 males, 74 females); the recapture probability was 79% (males) and 65% (females). Several multiple recaptures were obtained, one male individual was recaptured eight times, one female seven times. The maximum observed lifespan (*i.e.*, the longest time between first and last capture) was 11 days for males and 13 days for females (Table S1).

In the second generation (28 capture days; 11 July–13 August 2021), we marked 296 individuals (191 males, 105 females). 139 individuals (100 males, 39 females) were recaptured at least once, resulting in a recapture probability of 52% (males) and 37% (females). Sixty-one individuals (47 males, 14 females) had multiple recaptures, with three males being recaptured four times and two females even five times. The maximum lifespan was ten days for males and 13 days for females (Table S1).

### Population demography

The best model for the first generation presumed an additive effect of sex and quadratic time on the survival rate, an additive effect of sex and factorial time on the capture probability, an interactive effect of sex and squared time on the proportional recruitment pent, and an effect of sex on the number of individuals (Table 1(A)). This model estimated a population size of 155 males ($\pm$ 6 SE; *i.e.,* 15/ha $\pm$ 0.6 SE) and 155 females ($\pm$ 10 SE; *i.e.,* 15/ha $\pm$ 1.0 SE), which corresponds to a sex-ratio of 1:1.

During the first three capture days of the second generation, we only recorded one male specimen per day; these initial days were inappropriate for the applied algorithm and therefore were excluded from analyses due to high standard errors. The differences in the Akaike Information Criterion for the first four models were <2, and therefore a weighted average of the estimated population sizes over the best four models was used. The best four

**Table 1** Comparison of the best models of the POPAN 5.0 analyses for estimating the daily population sizes of *Boloria selene* in the (A) first and (B) second generation.

| Model | QAICc | Delta QAICc | QAICc weight | Model likelihood | No. Par |
|---|---|---|---|---|---|
| **(A) First generation** | | | | | |
| Phi(g+T$^2$) p(g+t) pent(g*T$^2$) N(g) | 2,849.82 | 0.0 | 0.60 | 1.0 | 34 |
| Phi(g+T) p(g+t) pent(g*T$^2$) N(g) | 2,852.41 | 2.6 | 0.165 | 0.27 | 33 |
| Phi(g+T$^2$) p(g+t) pent(g*T$^2$) N(.) | 2,852.5 | 2.7 | 0.158 | 0.26 | 33 |
| Phi(g+T) p(g+t) pent(g*T$^2$) N(.) | 2,853.96 | 4.1 | 0.076 | 0.13 | 32 |
| Phi(g+T$^2$) p(g*t) pent(g*T$^2$) N(g) | 2,862.87 | 13.0 | 0.0009 | 0.0015 | 54 |
| **(B) Second generation** | | | | | |
| Phi(g+T$^2$) p(g*t) pent(t) N(.) | 3,511.69 | 0.0 | 0.27 | 1.00 | 79 |
| Phi(g+T) p(g*t) pent(t) N(.) | 3,511.75 | 0.06 | 0.26 | 0.97 | 78 |
| Phi(T$^2$) p(g*t) pent(t) N(g) | 3,513.38 | 1.69 | 0.12 | 0.43 | 79 |
| Phi(T) p(g*t) pent(t) N(g) | 3,513.51 | 1.82 | 0.11 | 0.40 | 78 |
| Phi(g+T$^2$) p(g*t) pent(t) N(g) | 3,513.94 | 2.25 | 0.09 | 0.32 | 80 |

models share an effect of sex and factorial time on the capture probability and an effect of factorial time on the recruitment. The two best models assume a constant population size and an effect of sex and linear/squared time on the survival rate. In the third and fourth models, the number of individuals differs by sex and the survival probability is affected by time (squared or linear) (Table 1(B)). The calculated ĉ factor for under-dispersion was 0.41. The weighted model estimated a population size of 382 males ($\pm$ 18 SE; *i.e.,* 37/ha $\pm$ 4 SE) and 276 females ($\pm$ 19 SE; *i.e.,* 26/ha $\pm$ 3 SE), which corresponds to a sex ratio of 1.4 males per female.

The first generation displayed protandry, which was absent in the second generation (Fig. 2). The weather conditions were sufficient for flight activity throughout both flight periods and started to deteriorate at the end of the second flight period.

## Behaviour

Males were highly flight active in both generations (Fig. 3). The proportion of basking individuals compared to resting ones was significantly higher in the first than in the second generation (Chi-squared test of homogeneity: $\chi^2 = 11.87$, *df* $= 4$, $p = 0.018$). Females of the first generation were significantly more flight active than those of the second generation during which flight activity decreased while nectaring and resting increased (Chi-squared test of homogeneity: $\chi^2 = 15.47$, *df* $= 5$, $p = 0.009$). There were no significant differences in activity between sexes of the first generation (Chi-square test of homogeneity: $\chi^2 = 8.78$, *df* $= 4$, $p = 0.07$; N males $= 123$; N females $= 110$), but the difference was significant in the second generation (Chi-square test of homogeneity: $\chi^2 = 29,5$, *df* $= 5$, $p = 0.0001$; N males $= 191$; N females $= 105$).

## Plant genera used for nectaring

The first generation used nine different plant genera for nectaring, with *Knautia* as the most frequently used (males: 45.5%; females: 58.6%), followed by *Cardamine* for males (33.3%)

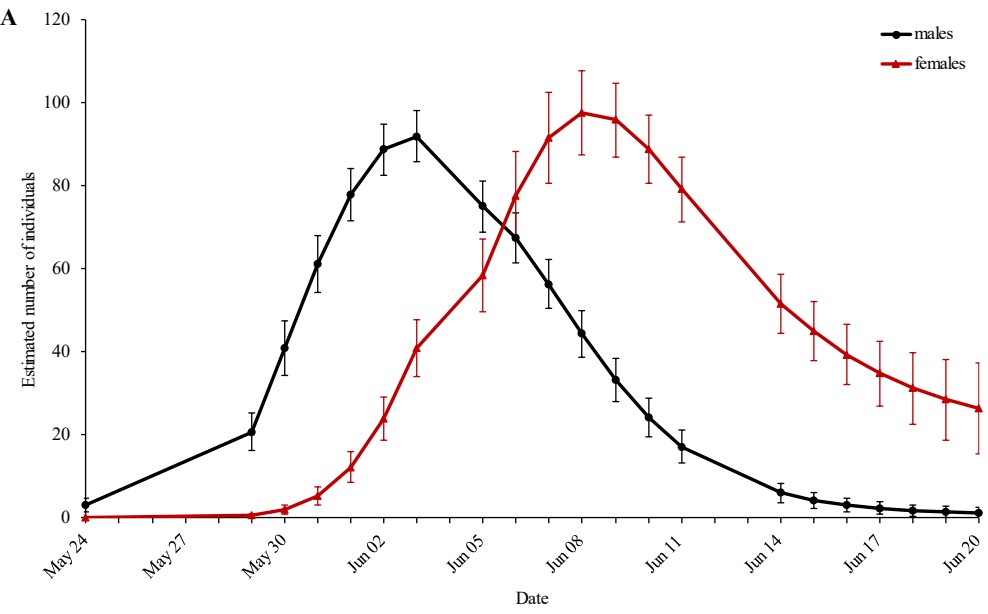

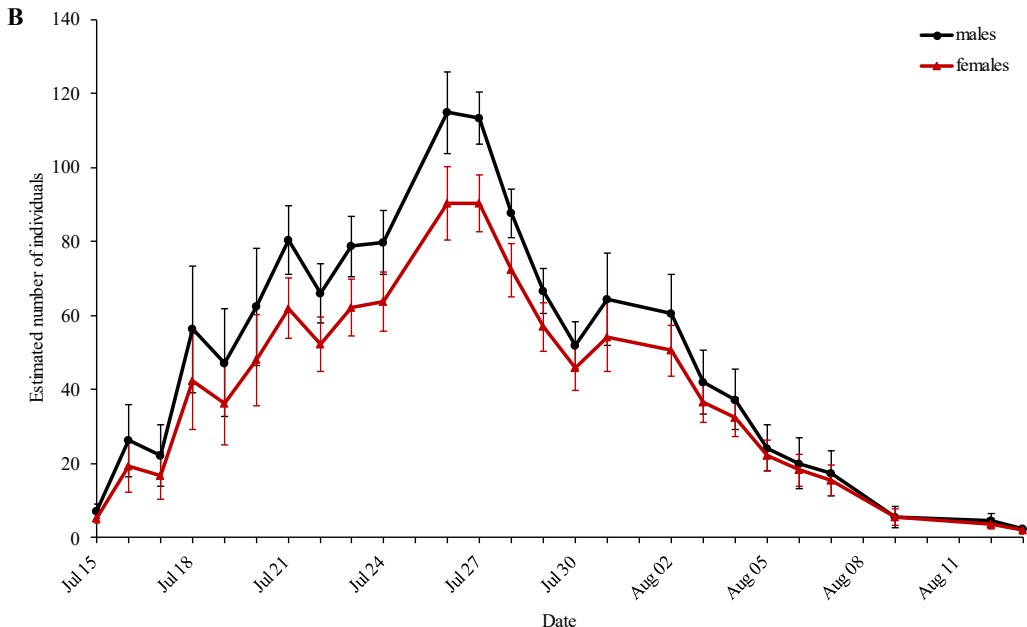

**Figure 2** **Estimated population size of the (A) first and (B) second generation of *Boloria selene* for every sampling day.** Black: males; red: females. Standard errors are given for all values.

and *Lychnis* for females (17.2%) (Fig. 4A). Three plant genera were used for nectaring only by males (*Arabidopsis*, *Veronica*, *Hieracium*), and another three only by females (*Myosotis*, *Ranunculus*, *Cirsium*), resulting in significant sex-specific preferences ($\chi^2 = 17.8$, $df = 8$, $p = 0.03$) (N males = 33; N females = 29). Removing all individuals recorded more than

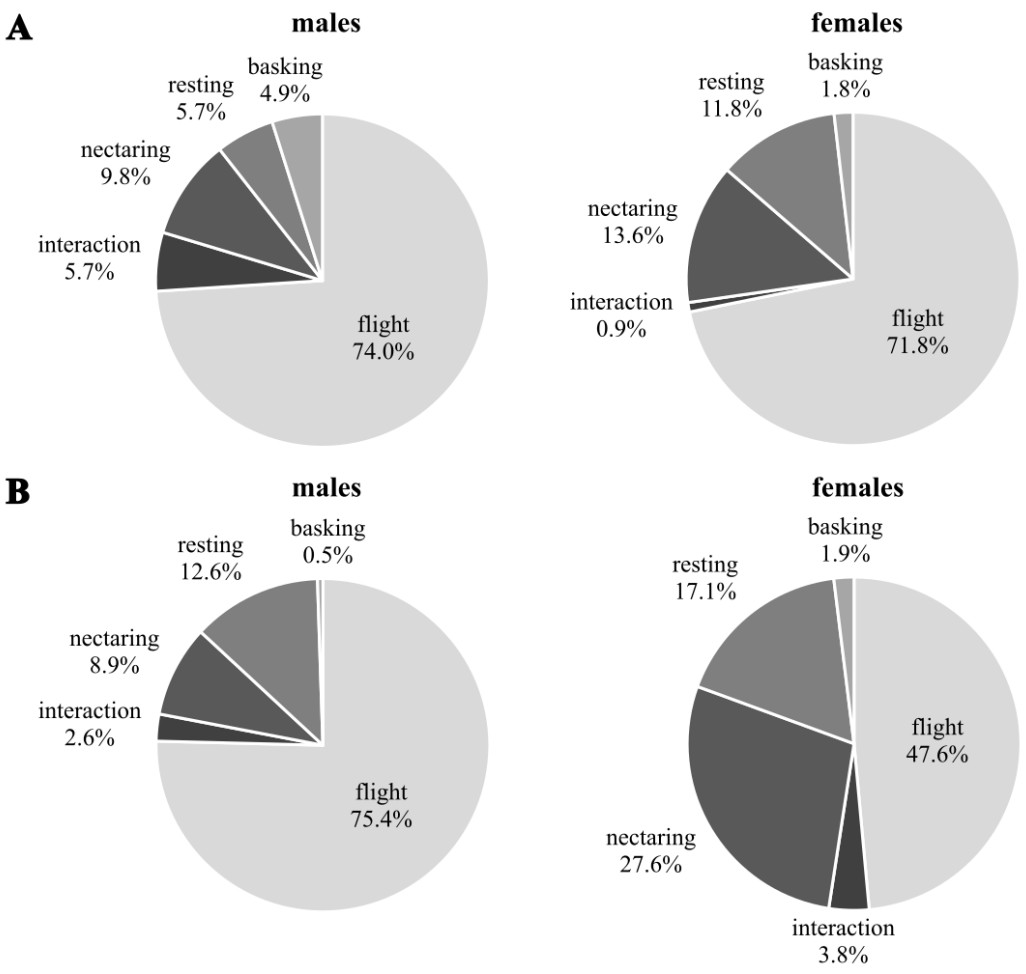

**Figure 3** Behaviour [%] observed during the first capture event of *Boloria selene* for males (left) and females (right) in the first (A) and second (B) generation.

once; however, the difference is only marginally significant ($\chi^2 = 13.6$, $df = 7$, $p = 0.06$) (N males = 28; N females = 23).

Seventy-four nectaring events were recorded in the second generation (27 males, 47 females). *Lythrum* (males: 44.4%; females: 46.8%) and *Cirsium* (males: 29.6%; females: 21.3%) (Fig. 4B) were visited most by both sexes. Despite both sexes using some genera exclusively, sex-specific preferences were not significant ($\chi^2 = 9.1$, $df = 10$, $p = 0.52$) (N males = 27; N females = 47).

## Wing condition

In the first generation, mean wing wear is linearly correlated with time, for males and females (LM males: $df = 11$, $R^2 = 0.91$, $p < 0.0001$, $N = 13$; LM females: $df = 12$, $R^2 = 0.85$, $p < 0.0001$, $N = 14$) (Fig. 5A). Mean wing wear did not differ significantly between sexes ($U$ test: $p = 0.56$). The slope of the correlation also did not differ significantly between

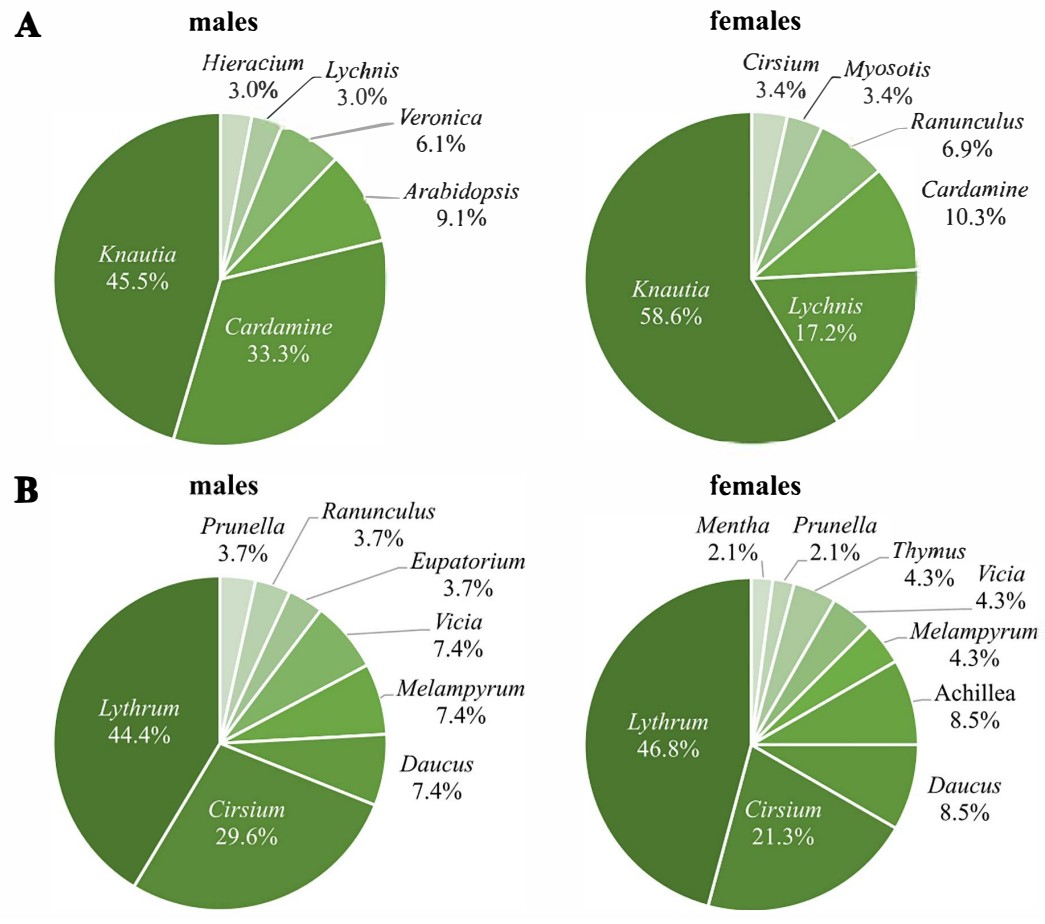

**Figure 4** Percentages of visited plant genera by *Boloria selene* males (left) and females (right) of the first (A) and the second (B) generation.

sexes (males: 0.09, females: 0.086), tested with a linear model with an interaction between the two predictors time and sex (F statistics: $df = 23$, $p = 0.79$).

In the second generation, the mean value of wing condition was linearly correlated with time for males (LM: $R^2 = 0.54$, $df = 13$, $p = 0.002$) but not for females (LM: $R^2 = 0.02$, $df = 11$, $p = 0.63$) (Fig. 5B). Means did not differ between sexes ($U$ test: $p = 0.43$), but the correlation slopes differed significantly (males: 0.03, females: 0.004; F-statistics: $df = 24$, $p = 0.037$). Comparing the slopes of wing decay of first and second generation, they were significantly steeper in the first (F statistics – males: $df = 24$, $p < 0.0001$; females: $df = 23$, $p < 0.0001$).

## Mobility and movement pattern

In the first generation, mean flight distance was significantly higher for females than for males (means – males: 99 m $\pm$ 112 SD, $N = 101$, females: 158 m $\pm$ 173 SD, $N = 71$) ($U$-test: $p = 0.005$) (Table 2). The mean lifespan movements were significantly higher in females as well ($U$ test: $p = 0.015$) (Table S1). A linear model with the number of days

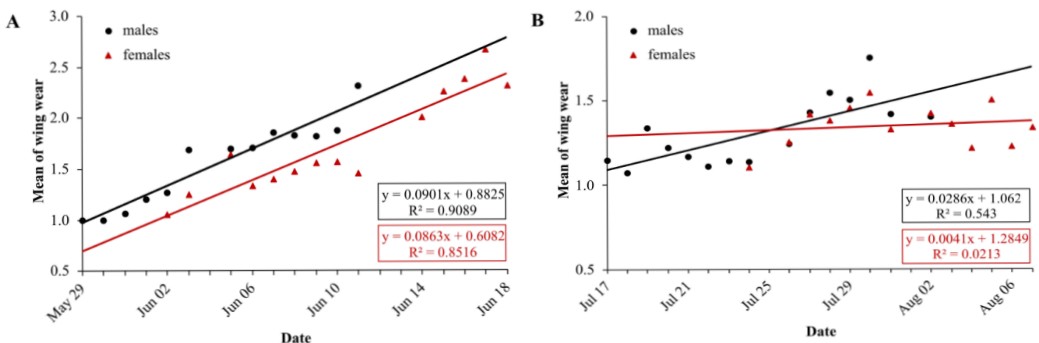

**Figure 5 Daily average wing condition of males (black) and females (red) of *Boloria selene* in the first (A) and second (B) generation.** Linear regression with formula as black line for males and red line for females; days with less than five individuals were excluded for the respective sex.

**Table 2 Minimum, maximum and mean distances between capture and first recapture for *Boloria selene* in the first and the second generation.**

|  | Generation 1 | | | Generation 2 | | |
|---|---|---|---|---|---|---|
|  | males | females | total | males | females | total |
| Min. distance (m) | 0 | 10 |  | 5 | 5 |  |
| Max. distance (m) | 602 | 919 |  | 1,070 | 812 |  |
| Mean (m) ± SD | 99 ± 112 | 158 ± 173 | 124 ± 143 | 97 ± 128 | 142 ± 190 | 109 ± 149 |
| Median (m) | 61 | 103 | 69 | 59 | 61 | 60 |
| Mean no. of days | 1.8 | 2.8 | 2.3 | 2.3 | 2.5 | 2.3 |

between the capture events as predictor and the flight distance as response variable did not reveal a correlation for neither sex (males: $df = 99$, $R^2 = 0.0002$, $p = 0.9$; females: $df = 69$, $R^2 = 0.006$, $p = 0.5$) (Fig. S1A).

In the second generation, mean flight distance was again higher for females (142 m ± 190 SD, $N = 39$) than for males (97 m ± 128 SD, $N = 100$) (Table 2), but these did not differ significantly ($U$ test: $p = 0.9$), nor did the lifespan movements ($U$ test: $p = 0.4$) (Table S1). There was no significant correlation between passed time since capture and flight distance (males: $df = 98$, $R^2 = 0.03$, $p = 0.08$; females: $df = 37$, $R^2 = 0.006$, $p = 0.17$) (Fig. S1B).

The travelled distances did not differ significantly between generations, neither for males ($U$ test: $p = 0.9$), nor for females ($U$ test: $p = 0.1$).

In the extrapolation of dispersal potential (Fig. 6), the best fits of NEF and IPF were obtained for 50 m classes; only NEF for first-generation males had its best fit with 20 m interval. In the first generation, NEF always had a better fit than IPF, while it was the opposite in the second generation (Table 3). In both generations, the predicted dispersal probabilities were always higher for IPF than for NEF, and for females than for males. Comparing generations, the modelled proportions of long-distance movements were

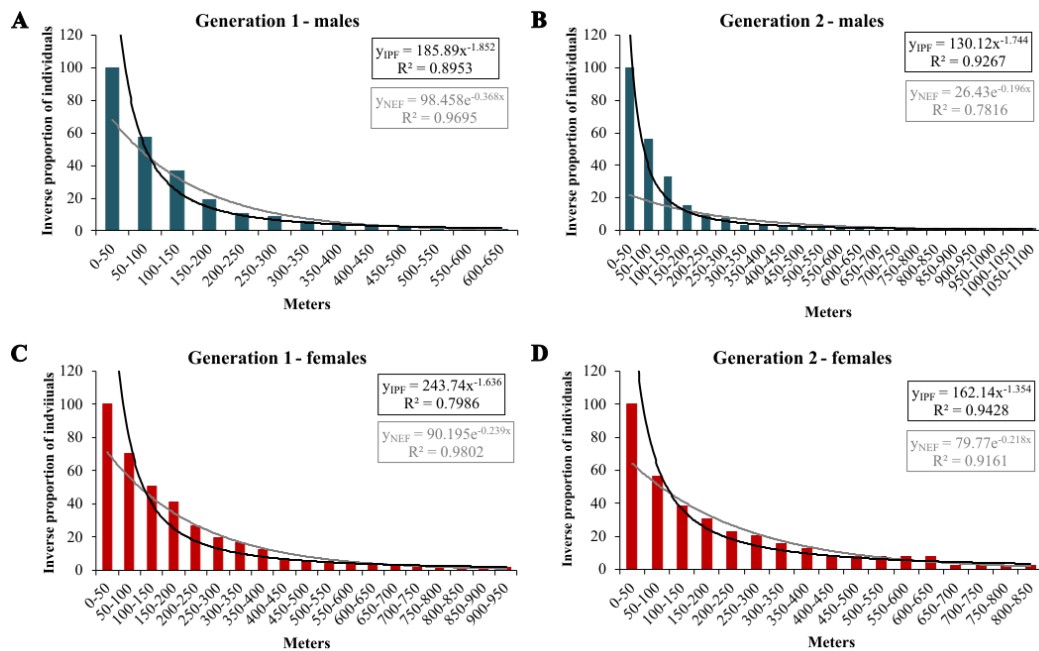

**Figure 6** Inverse proportion of individuals reaching certain distance classes in 50 m intervals for *Boloria selene* with the fitted NEF (grey lines) and IPF (black line). Males (A, B; blue) and females (C, D; red) in the first (A, C) and second (B, D) generation.

**Table 3** Stability index ($R^2$) for NEF and IPF based on 20, 30, and 50 m intervals for the first and the second generation of *Boloria selene*; highest values for each function are in bold.

|  | 20 m intervals | | 30 m intervals | | 50 m intervals | |
|---|---|---|---|---|---|---|
|  | **IPF** | **NEF** | **IPF** | **NEF** | **IPF** | **NEF** |
| **Generation 1** | | | | | | |
| males | 0.717 | **0.970** | 0.806 | 0.969 | **0.895** | 0.969 |
| females | 0.621 | 0.965 | 0.706 | 0.966 | **0.799** | **0.980** |
| **Generation 2** | | | | | | |
| males | 0.716 | 0.762 | 0.815 | 0.771 | **0.927** | **0.782** |
| females | 0.824 | 0.910 | 0.881 | 0.910 | **0.943** | **0.916** |

considerably higher in the second generation; except for the IPF of first-generation males (Table 4).

The flight paths in both generations showed long and short flights for both sexes and no sex-specific movement pattern (Figs. S3 & S4). Accumulations of butterfly individuals were in the majority of cases in close proximity to mapped stands of the larval food plant *Viola palustris* but accumulations also existed without the confirmed presence of this plant (Fig. S5).

**Table 4** Percentages of individuals of *Boloria selene* in the first and the second generation assumed to travel 1, 2, 3 or 5 km calculated with the best fitting NEF and IPF and the corresponding distance intervals in brackets.

| Generation 1 | males | | females | |
|---|---|---|---|---|
| | NEF (20 m) | IPF (50 m) | NEF (50 m) | IPF (50 m) |
| **Generation 1** | | | | |
| 1 km | 0.06 | 0.85 | 0.76 | 1.81 |
| 2 km | $3.85 \times 10^{-5}$ | 0.29 | $6.00 \times 10^{-3}$ | 0.58 |
| 3 km | $2.40 \times 10^{-8}$ | 0.16 | $5.30 \times 10^{-5}$ | 0.30 |
| 5 km | $9.30 \times 10^{-15}$ | 0.07 | $3.80 \times 10^{-9}$ | 0.13 |
| **Generation 2** | NEF (50 m) | IPF (50 m) | NEF (50 m) | IPF (50 m) |
| 1 km | 0.52 | 0.70 | 1.02 | 2.81 |
| 2 km | 0.01 | 0.21 | 0.01 | 1.10 |
| 3 km | $2.06 \times 10^{-4}$ | 0.10 | $2.00 \times 10^{-4}$ | 0.63 |
| 5 km | $8.13 \times 10^{-8}$ | 0.04 | $2.72 \times 10^{-8}$ | 0.32 |

## DISCUSSION

### Population structure and protandry

The observed population density of both generations had intermediate levels if compared to other studies (*Cozzi, Müller & Krauss, 2008*; *Konvicka et al., 2011*) ; this indicates suitable habitat quality. The higher density of the second generation might be explained by an increased survival rate of the non-overwintering larvae which is a common feature in bi- and multivoltine species (*Altermatt, 2010*; *Plazio & Nowicki, 2021*). The overall moderate population size of both generations facilitated high recapture rates (*Fischer & Fiedler, 2001b*; *Fric, Klimova & Konvicka, 2006*). Sex-specific differences in the recapture rates between generations might be explained by the decreased flight activity of second-generation females (*Weyer & Schmitt, 2013*; *Jugovic, Crne & Luznik, 2017*).

In the majority of cases, protandry is assumed to be beneficial for the reproductive success of both sexes (*Fagerström & Wiklund, 1982*; *Zonneveld, 1992*) and is achieved by faster development and earlier eclosion of males (*Wiklund, Nylin & Forsberg, 1991*; *Fischer & Fiedler, 2000*). However, this beneficial feature was only observed for the first and not for the second generation. This raises the question why this apparently advantageous trait is not always expressed. We assume that the first generation of *Boloria selene* is synchronized when winter diapause ends in early spring. During the flight period of the first generation, the eggs are laid along a two-week time window; additionally, the second generation apparently is lacking a synchronizing environmental cue. This results in asynchronous eclosion of the second generation, blurring protandry (or at least weakening it so strongly that it could not be detected by our model) and making the second-generation flight period longer than the one of the first, apparently also affecting the time period of egg-laying (*Komata & Sota, 2017*). A prolongation of the second generation due to bad weather conditions that might have limited the daily flight periods (*Wiklund, Nylin & Forsberg, 1991*; *Junker & Schmitt, 2010*; *Larsdotter Mellström & Wiklund, 2010*), is rather unlikely because weather conditions were rather favourable all along the second flight period of *Boloria selene*.

## Behaviour and use of nectar plants

First-generation males and females had similar activity patterns. This is a rather unusual pattern for butterflies (but see *Parile, Piccini & Bonelli, 2021*) because males in general fly more whereas females rest and feed more (*Zimmermann et al., 2005*; *Ehl et al., 2019*). Interestingly, this pattern changed during the second generation. While male flight activity remained unchanged, it decreased for females with an increase of nectaring, thus reflecting the commonly observed pattern. The analysis of wing condition decay supports these flight activity patterns, with sex-specific differences and a slower decay in females only for the second generation. The generally high male flight activity is explained by their search for mates; females on the other hand search for nectar sources and oviposition sites, activities not demanding such high investments in flight time (*Ebert & Rennwald, 1991*; *Fischer, Beinlich & Plachter, 1999*; *Kuras et al., 2003*; *Weyer & Schmitt, 2013*).

These sex- and generation-specific differences in behaviour also coincide with differences in nectar plant choice, with first-generation individuals being more selective about plant genera and exhibiting sex-dependent selection, while the second generation was mostly opportunistic. A general decrease in the availability of plants for nectaring in the late season might be one possible explanation for our results, supported by *Rusterholz & Erhardt (2000)*.

Flower selection by butterflies also depends on the specific nectar composition as males rely more on sucrose and sugar because of their higher, energy-demanding flight activity, while females have a higher demand for amino-acids for egg production (*Rusterholz & Erhardt, 2000*; *Mevi-Schütz & Erhardt, 2002*). Thus, the nectar sources available during the first generation might be more efficient to satisfy butterflies' nutritional needs, for males and females, so that individuals need less time for feeding. In particular, *Knautia* is a highly efficient nutritional source for both sexes because its nectar contains similar proportions of sucrose, glucose and fructose and high amounts of essential and non-essential amino-acids (*Venjakob, Leonhardt & Klein, 2020*). When such high-quality resources are missing during the second generation, individuals are forced to use a mix of different plants and to visit more inflorescences to get the same amount of nutrition, probably explaining the larger time investment of second-generation females in nectaring. Interestingly, females broadened their nectaring behaviour and their time investment considerably more than males. This might underline the higher difficulty of females fulfilling their nutritional needs compared with males.

Furthermore, *Rusterholz & Erhardt (2000)* observed that females of *Polyommatus bellargus* were more selective for nectar sources with high amino-acid content in the spring than in the autumn generation. They hypothesised that overwintering negatively affects the nutritional conditions of the larvae and consequently enhances the requirements for amino-acids of spring-generation females. This might also explain the broader range of nectar sources used in our second generations and the smaller numbers in the first generation because individuals of the first generations might be forced to be more selective for the most nutrient-rich nectar plants. The negative effect of overwintering on the nutritional conditions of larvae might also in males lead to more specialised nectar and nutrient requirements after eclosion as males have a generally higher weight loss caused by

metamorphosis, as shown for *Lycaena tityrus* (*Fischer & Fiedler, 2000*). The hypothesis of a seasonal change in nectar plant selectivity might therefore not only be true for females, as supposed by *Rusterholz & Erhardt (2000)*, but also for males and well corresponds with our observations.

## Movement patterns and dispersal

First-generation males had a slightly higher likelihood of emigrating than those of the second generation if considering IPF extrapolations as suggested by *Baguette (2003)*. For females, we observed the opposite with considerably higher extrapolations for long-distance dispersal in the second than in the first generation. While males are motivated by the search for females and thus respond with emigration to lower population densities (*Kuussaari et al., 1998*), female mobility is affected by the availability of suitable oviposition host plants after mating (*Timus et al., 2017*). Consequently, high population densities increase intraspecific competition for host plants, hereby enhancing dispersal. Additionally, male harassment is augmenting with higher population densities and therefore may also lead to increased emigration pressure after mating (*Baguette et al., 1998*).

This is well in concordance with our findings for *Boloria selene*. The species' sex- and generation-specific differences in dispersal probabilities can therefore be explained by the different densities of the two generations and the different evolutionary functions of both sexes, maybe even mirrored in their morphology as suggested by work of *Fric, Klimova & Konvicka (2006)* obtaining significant differences in body design between generations. The overall higher dispersal probability of females compared to males (in particular in the second generation) can be explained by the sex-specific advantage of potential founder effects: After mating, one individual female can establish a new population in a so-far empty habitat, whereas the reproductive success of males always depends on the presence of females (*Weyer & Schmitt, 2013*).

## CONCLUSIONS

The differences between both generations and sexes of *Boloria selene* most likely are explained by adaptation to seasonal selection pressures and different functions of sexes to optimise respective reproductive success. The intersexual differences included behaviour, average flight distances, plants used for nectaring, and the probability of long-distance movements. Thus, males fly more constantly to find mating partners, females fly less often but longer distances, searching for nectar sources and oviposition sites. These behavioural differences might influence the need for specific nectar ingredients as males demand sugar to maintain their energy-consuming flights and females need amino-acids to produce eggs. Winter diapause apparently acts as a synchronising event that occurs as a prerequisite for protandry in the first generation. Associated with overwintering is an increased mortality, which results in lower first-generation densities. The second generation lacks such a synchronisation event, eliminating the reproductive advantage of protandry. The absence of diapause in combination with the more favourable conditions of the summer season result in higher survival rates in the offspring, und thus in higher population density. This higher density is also impacting dispersal, being reduced in males due to higher densities of

mating partners and enhanced in females due to higher competition for suitable oviposition sites.

## ACKNOWLEDGEMENTS

We thank Clara Bauer (FÖJ) as well as several interns and students of the University of Potsdam for assistance during field work. We are grateful to Zdeněk Faltýnek Fric and one anonymous reviewer for their constructive comments on a previous version of this publication.

### Funding

This work had no additional external funding.

### Competing Interests

The authors declare there are no competing interests.

### Author Contributions

- Ann-Kathrin Sing performed the experiments, analyzed the data, prepared figures and/or tables, authored or reviewed drafts of the article, and approved the final draft.
- Laura Guderjan performed the experiments, authored or reviewed drafts of the article, and approved the final draft.
- Klara Lemke performed the experiments, authored or reviewed drafts of the article, and approved the final draft.
- Martin Wiemers conceived and designed the experiments, authored or reviewed drafts of the article, and approved the final draft.
- Thomas Schmitt conceived and designed the experiments, authored or reviewed drafts of the article, and approved the final draft.
- Martin Wendt analyzed the data, authored or reviewed drafts of the article, and approved the final draft.

### Field Study Permissions

The following information was supplied relating to field study approvals (i.e., approving body and any reference numbers):

Field experiments were approved by Naturschutzbehörde Frankfurt/Oder.

### Data Availability

The raw data are available in the Supplemental Files.

### Supplemental Information

Supplemental information for this article can be found online at http://dx.doi.org/10.7717/peerj.16965#supplemental-information.

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
