# Peer review of "Different ecological demands shape differences in population structure and behaviour among the two generations of the small pearl-bordered fritillary"

_PeerJ, doi:10.7717/peerj.16965_

## Round 0.1 · original submission · Minor Revisions

Dear Dr. Schmitt,

After this first review round, both reviewers indicated the need for minor reviews in your manuscript! Well done!

Please prepare a new version of your manuscript and a rebuttal letter informing the changes that were made and those that were not accepted by you. Congratulations!

Sincerely,
Daniel Silva

Reviewer 1 ·

Basic reporting

An interesting paper, well documented, clearly and correctly written.
For a better understanding by the readers I suggest two small additions.
Lines 71-73 it would be good to mention that also in the case of the genus Boloria, the 2nd generation can only be partial, a situation in which the number of individuals is smaller than in the first generation (Klimczuk & Sielezniew, 2020).
Lines141-143 it is worth mentioning that Boloria selene is bivoltine in temperate areas below 1000m altitude, but the 2nd generation can sometimes be only partial (Schweizerischer Bund für Naturschutz, 1987). At altitudes above 1000m as well as in northern Europe, it is univoltine.

Experimental design

The field data and methodology used are correct. The statistical processing, discussions and conclusions are adequate and appropriate to the objectives of the paper.

Validity of the findings

The paper is original and makes many new contributions to the biology and ecology of Boloria selene.

Additional comments

For a better understanding by the readers I suggest two small additions.
Lines 71-73 it would be good to mention that also in the case of the genus Boloria, the 2nd generation can only be partial, a situation in which the number of individuals is smaller than in the first generation (Klimczuk & Sielezniew, 2020).
Lines141-143 it is worth mentioning that Boloria selene is bivoltine in temperate areas below 1000m altitude, but the 2nd generation can sometimes be only partial (Schweizerischer Bund für Naturschutz, 1987). At altitudes above 1000m as well as in northern Europe, it is univoltine.

·

Basic reporting

The topic of the paper is to study demographic parameters of Boloria selene, a bivoltine species of butterfly from the family Nymphalidae, using Mark-Release-Recapture methods. The study is well-designed, all calculations are appropriate and the results are sound and interesting. The paper is great and I really like its outputs.I would only like to add several things as we also studied this and other butterfly species in the past.
First of all, this is not a first MRR study of a vivoltine butterfly species. I am aware of two our studies, one about Araschnia levana (see Fric & Konvicka 2000, Nota Lepid) and another with several species of Melitaea, where one was bivoltine Melitaea didyma (see Hajkova et al. 2023, J. Nat. Cons). Also we used Boloria selene in a paper dealing with a biomechanics pattern (Fric et al. 2006, Evol. Ec. Res.), where we found significant differences in body design between spring and summer generations of this species. I guess that the last paper is relevant to a discussion about different flight performances of this species?
Howeve, I am not writing this a a self-promotion!

Experimental design

Experimental design is appropriate for this type od study. It requires a high effort, but with it, it is possible to obtain a lot of results. The description os sufficient.

Validity of the findings

The conclusions are well stated and the results are solid and interesting.

Additional comments

I have no further comments.

---

## Round 0.2 · accepted · Accept

Dear Dr. Schmitt,

I am pleased to inform you that your manuscript has just been accepted for publication in PeerJ. Congratulations!

Sincerely,
Daniel Silva

·

Basic reporting

The authors studied population demography parameters in two generations of butterfly Boloria selene. This is the second round of review, in the first round, I already wrote that the manuscript is clear and well structured, with interesting results. I suggested only minute modifications, and the authors applied them. I have no further comment.

Experimental design

The experimental design is correct, as well as the following data collection and analyses.

Validity of the findings

The findings are strong and the findings are well stated, linked to original research questions.

Additional comments

The authors answered all my concerns in their manuscript and I believe that it is ready for a publication